# NFkB Pathway and Hodgkin Lymphoma

**DOI:** 10.3390/biomedicines10092153

**Published:** 2022-09-01

**Authors:** Fabrice Jardin

**Affiliations:** INSERM U1245, Henri Becquerel Center, IRIB, Normandy University, 76000 Rouen, France; fabrice.jardin@chb.unicancer.fr

**Keywords:** Hodgkin lymphoma, NFkB pathway, cfDNA, targeted therapy

## Abstract

The tumor cells that drive classical Hodgkin lymphoma (cHL), namely, Hodgkin and Reed-Sternberg (HRS) cells, display hallmark features that include their rareness in contrast with an extensive and rich reactive microenvironment, their loss of B-cell phenotype markers, their immune escape capacity, and the activation of several key biological pathways, including the constitutive activation of the NFkB pathway. Both canonical and alternative pathways are deregulated by genetic alterations of their components or regulators, EBV infection and interaction with the microenvironment through multiple receptors, including CD30, CD40, BAFF, RANK and BCMA. Therefore, NFkB target genes are involved in apoptosis, cell proliferation, JAK/STAT pathway activation, B-cell marker expression loss, cellular interaction and a positive NFkB feedback loop. Targeting this complex pathway directly (NIK inhibitors) or indirectly (PIM, BTK or NOTCH) remains a challenge with potential therapeutic relevance. Nodular predominant HL (NLPHL), a distinct and rare HL subtype, shows a strong NFkB activity signature because of mechanisms that differ from those observed in cHL, which is discussed in this review.

## 1. Introduction

Hodgkin lymphoma (HL) is a well-defined B-cell malignancy that encompasses two main subtypes, classical HL (cHL), which accounts for 95% of cases, and nodular lymphocyte predominant HL (NLPHL). cHL is one of the most common lymphomas, with an incidence of 3/100,000 new cases per year in Western countries. This lymphoma has a bimodal distribution according to age, preferentially affecting adolescents or young adults and those over 60 [1]. It is a disease with a generally good prognosis that has benefitted from the many advances in immunotherapy, notably the use of anti-CD30 antibodies and immune checkpoint inhibitors (ICI), after a period based exclusively on chemotherapy and radiotherapy [2]. The tumor cells that drive this neoplasia, namely, Hodgkin and Reed-Sternberg (HRS) cells, display hallmark features that include their rareness in contrast with an extensive and rich reactive microenvironment, their loss of B-cell phenotype markers, their immune escape capacity and the activation of several key biological pathways, including the JAK/STAT signaling, MAPK/ERK and NFK-B pathways [3,4]. Despite the difficulties in isolating and studying HRS cells in vivo, both canonical and noncanonical NFkB appear to be activated, indicating a crucial role of this pathway in HL pathogenesis (Figure 1) [2,3,5]. Here, we provide an overview of the mechanisms and consequences of NFkB pathway dysregulation in HL and their potential therapeutic relevance.

## 2. Constitutive Activation of the NFkB Pathway

HRS cells have been demonstrated to express, primarily in cell lines, the five NFkB transcription factors. Indeed, nuclear REL, p50 and p65 were identified in HRS cell lines as p50-p65 and p50-REL heterodimers and p50-p50 homodimers. TRAFs were also identified, demonstrating the activity of the canonical NFkB pathway [6,7]. In addition, Bcl-3, an inducer of nuclear p50-p50, is associated with this dimer in HRS cell lines [8].

The alternative NFkB pathway is also similarly activated in HRS cells, as shown by the detection of p52-RelB heterodimers and NIK protein in both cell lines and primary HRS cells [3,5,9,10]. For instance, patient biopsies showed stable NIK protein expression in 49/50 cHL cases, indicating that NIK and the noncanonical pathway are highly prevalent in cHL [9]. However, this molecular feature is not specific, and is shared by closed entities, such as primary mediastinal B-cell lymphoma (PMBL) and mediastinal gray zone lymphoma (MGZL) [11]. Importantly, the inhibition of both canonical and noncanonical NFkB activity in HL cell lines is toxic for these cells, causing reduced proliferation and increased apoptosis, which demonstrates the essential role of the constitutive activation of both NFkB pathways for HRS cell survival [12,13].

The transcriptional consequence of NFkB pathway activation in cHL has been partially recognized. The molecular signature of NFkB-related genes distinctly differs between cHL and other lymphoid malignancies, but correlates with that of normal GC B cells [10]. In this setting, the canonical and noncanonical NFkB dimers regulate common and distinct gene sets, and if most target genes are upregulated by NFkB, the expression of a fraction of target genes is inhibited by NFkB [10].

The cell survival of HRS cells appears to be predominantly controlled by the noncanonical NFkB pathway, involving the upregulation of proteins that block both intrinsic and extrinsic pathways. The canonical pathway, via p50-RelA, contributes to survival via the upregulation of BCL-XL [10]. Notably, NFkB inhibition resulted in marked spontaneous and p53-independent apoptosis, which could be rescued by the ectopic expression of BCL-XL, underscoring its dominant role in the survival of H/RS cells [14]. Conversely, BCL2 is exclusively controlled by Rel-B. Both mRNA and protein levels decrease following Rel-B depletion, and exogenous BCL2 expression partially rescue the death induced by decreased Rel-B in HRS cells [15].

In addition to apoptosis regulation, NFkB pathway activation leads to the expression of several genes coding for chemokines (including CCL5, CCL17, CCL22 and CCR7), cytokines, receptors, apoptotic regulators, intracellular signaling molecules and transcription factors. Indeed, 45 genes have been recognized in two cell lines to be regulated by NFkB. Chromatin immunoprecipitation experiments have shown that NFkB factors are recruited directly to the promoters of several target genes, including signal transducer and activator of transcription (STAT)5a, interleukin-13 and CCR7. Finally, NFkB activity in HL cell lines reduced the expression of several B-cell factors, including BCL6, PAX5 and FOXP1, and therefore may contribute to the loss of the B-cell phenotype of HRS cells [10].

## 3. Activation of the NFK-B Pathway, the First Hit?

The primary causes of the emergence of cHL disease are not known. Some etiological or facilitating factors have been suggested, and may be considered to be the first steps in a complex process involving the early activation of the NFkB pathway.

Approximately 30–50% of cHL cases are Epstein Barr virus positive (EBV+), and this endemic virus constitutes the best-established etiological factor in cHL. EBV+ HRS cells display a latency-type profile defined by the expression of EBV nuclear antigen 1 (EBNA1) and latent membrane proteins 1 (LMP1) and 2a (LMP2a). EBNA1 is implicated in viral replication by maintaining the viral episome in proliferating cells. LMP2a contains an ITAM motif, a key element for BCR signaling, mimicking the presence and activity of a BCR absent in HRS cells. LMP1 carries in its cytoplasmic domain two carboxy-terminal activating regions (CTAR) that interact with the TRAFs molecule to mimic a constitutively active CD40 receptor, which consequently activates both canonical and noncanonical pathways [16,17,18] (Figure 2).

25(OH) Vitamin D (vitD) deficiency has also been identified as a potential facilitating factor for the emergence of cHL. If the well-known primary function of VitD is to control calcium metabolism, VitD has also been suggested to be involved in the aggressiveness and outcome of cHL. In a gene expression profiling study, Donati and colleagues showed that VitD deficiency induced profound changes in the transcriptional program leading to the NFκB-mediated activation of stress-protective and pro-survival pathways. More specifically, thirty-one genes linked to the TLR/NF-κB pathway, in particular, TLR1, TLR4, TLR5, TLR6, TLR8 and TLR10, were found to be significantly upregulated in VitD-deficient cHL. Furthermore, the immediate downstream effectors of the canonical signaling of this pathway were also significantly upregulated, including MyD88, IRAK4 and TRAF6. In contrast, VitD deficiency does not significantly alter gene expression in diffuse large B-cell lymphoma (DLBCL), suggesting that the mechanism of action depends on the lymphoma subtype [19]. Consistent with this observation, VitD signaling, defined by vitamin D receptor (VDR) expression analysis, was highly activated in cHLs but not in DLBCLs. The role of vitD in this setting is also corroborated by the fact that VitD directly inhibits the NFκB signaling pathway through several mechanisms [19].

Another field of research for deciphering factors that may contribute to the emergence of cHL is based on the analysis of single nucleotide polymorphisms (SNPs) in genes encoding key physio-pathological proteins. GWASs of cHL have provided evidence that variations in a number of genes may influence the risk of developing HL. In a meta-analysis of seven genome-wide association studies totaling 5325 HL cases and 22,423 control patients, integration of gene expression, histone modification, and in situ promoter capture Hi-C data at the five new and 13 known risk loci implicates the dysfunction of the germinal center reaction, disrupted T-cell differentiation and function and constitutive NFκB activation as mechanisms of predisposition [20]. Although functional analyses are required to determine the biological basis of cHL association signals, GWAS analysis has demonstrated that these risk loci are enriched for regulatory elements in B cells, including RELA [21]. Some SNPs can also be involved in the clinical presentation and aggressiveness of the disease. For instance, a SNP in the intronic region of the NFkappaB1 gene (SNP/iNFKB1, rs1585215), an important regulator of cytokine gene expression, is more frequently associated with stage IV and extranodal disease at diagnosis [22].

## 4. Somatic Mutations Related to NFkB Pathways/Genetic Lesions of Components of the NFkB Pathway

Despite HRS cell paucity in biopsies, the landscape of somatic mutations based on the analysis of microdissected cells, HRS cell lines, flow cytometry sorting cells or cfDNA analysis currently provides an accurate description of the somatic mutation landscape [23].

Indeed, mutations/genomic alterations in members of the NFκB pathway are a main feature of HRS cells that sustain NFkB deregulation, and are reported in approximately 50% of cHL cases [24]. These alterations include both amplification/gain copies of activators and mutation/inactivation of the negative regulators of the canonical and noncanonical NFK-B pathways (Figure 2). Gains and amplifications of the genes encoding the NF-κB factor REL and the kinase MAP3K14 (NIK) have been reported, whereas *BCL3*, an unusual member of the IKB family, promotes NFkB activity by binding to p50 homodimers and is also rarely targeted by translocation cHL [25,26,27].

Conversely, inactivating mutations in the genes *TNFAIP*3, *NFKBIA* and *NFKBIE*, which encode negative NF-κB regulators, are more frequent [28,29,30,31]. Genetic lesions can also less frequently target the negative regulators CYCLD and TRAF3 [32]. These mutations are not specific to cHL and are shared by PMBL and MGZL, suggesting a common genetic basis of these close but distinct entities. In contrast, nonthymic GZL displays a divergent somatic mutation pattern, sparing the NFkB pathway component [11,33].

The NFkB inhibitor I-kappa-B-epsilon (NFKBIE) gene, which encodes IκBε, is targeted by a recurrent 4-bp truncating mutation in addition to rare point mutations. Interestingly, a high and similar frequency of NFKBIE aberrations is observed in PMBL and cHL [34]. In chronic lymphocytic leukemia (CLL), NFKBIE-mutated cells showed reduced IκBε protein levels and decreased p65 inhibition, along with increased phosphorylation and nuclear translocation of p65 [34,35]. The biological relevance of NFKBIE aberrations in cHL remains to be determined.

XPO1 is the main nuclear export molecule in eukaryotic cells. *XPO1* E571K is a mutation hotspot observed in 10–30% of cHL [36,37]. High expression of XPO1 increases the efflux of IkB, promoting its proteasomal degradation in the cytoplasm and resulting in higher NFkB activity. To date, the ability of the *XPO1* E571K variant to increase NFkB activity and whether its pharmacological targeting by SINE molecules is efficient in this setting have not been clearly demonstrated [37,38,39,40].

Finally, cfDNA analysis indicates that multiple mutations often target the NFkB component, which may suggest synergistic dysregulation of the NFkB pathway [30,41,42]. Notably, the pattern of somatic mutation is most likely influenced by the cellular context. Indeed, *TNFAIP3* and *NFKBIA* mutations are more frequent in EBV-uninfected cHL, indicating that in EBV-positive cases, viral LMP1, as a strong NFκB activator, can replace the need to inactivate *TNFAIP3* or *NFKBIA* [28,30].

## 5. HL Microenvironment and NFkB Activation

The microenvironment in cHL contains various cell types, including the most abundant CD4+ T cells, regulatory T cells (Treg), nonmalignant reactive B cells, mast cells, neutrophils, eosinophils, and stromal cells (Figure 1). These cells shape a supportive and pro-survival environment for HRS that in turn expresses many chemokines/cytokines and various receptors able to activate canonical and alternative NFK pathways.

HRS cells express several members of the TNF superfamily, including CD40, BCMA, TACI, and RANK. The so-called rosettes formed by CD4+ T cells surrounding HRS cells can visualize the interaction between CD4+ T cells and HRS [43,44]. This phenomenon involves CD40 (TNFRSF5) and CD40 L, inducing NFkB activation and a positive autocrine loop by increasing the expression of CD40 itself but also that of CD80, CD86, and apoptotic genes [45]. In EBV-positive HRS cells, the LMP1 protein mimics CD40 signal transduction and can lead to NF-κB activation [46]. CD30 L is expressed by mast cells and eosinophils and binds to CD30, which is associated with TNF receptor-activated factors (TRAFs), to activate both the canonical and alternative NFkB cascades [47]. APRIL and BAFF, expressed by neutrophils, bind to BCMA and TACI on HRS cells, which in turn activate the alternative NFkB pathway [48].

The amplification of PDL1 and PDL2 on tumor cells, coded on chromosome 9p24.1, is mediated by PIM serine/threonine kinases on tumor cells via the constitutive activation of the NFkB and JAK-STAT signaling pathways [49], indicating that NFkB contributes to the immune escape of HRS cells. HRS cells also express potent immunomodulatory proteins facilitating tumor immune escape, namely Gal-1, PD-L1, and PD-L2, at least in part via the increased activity of STAT and NFkB transcription factors [50].

Recently, the key role of lymphotoxin-a (LTA) as the causative factor in the autocrine and paracrine activation of canonical and noncanonical NFkB was revealed in cHL cell lines. LTA was isolated from cHL cell lines in the secretome after analysis by chromatography and subsequent mass spectrometry. Indeed, LTA was the only cytokine isolated from 72 extracellular proteins obtained from the culture supernatant of the 1236 cHL line. The addition of the supernatant or a recombinant LTA molecule leads to the activation of both canonical and noncanonical NFkB pathways in HeLa cells resembling the constitutive activity in cHL. Conversely, the addition of etanerecpt, a neutralizing decoy receptor of LTA, blocks NFkB activation. Furthermore, gene expression data from microdissected cells confirmed the high level of expression of LTA by HRS cells, as compared to other B-cell lymphomas. LTA and its receptor TNFRSF14 are transcriptionally activated by noncanonical NFkB, creating a continuous feedback loop. Experiments indicate that LTA shapes the expression of cytokines, receptors, immune checkpoint ligands, and adhesion molecules, including CSF2, CD40, PD-L1/PD-L2, and VCAM1. Together, these results lead to the conclusion that LTA is a crucial inducer of the NFkB pathway in cHL [51].

## 6. Targeting the NFkB Pathway in cHL

Given the crucial role of the NFkB pathway in the pathophysiology of HL and its involvement in the survival mechanisms of HRS cells, considering the therapeutic and pharmacological targeting of this pathway seems appropriate.

In early-stage cHL, the standard of care is the combination of multidrug chemotherapy (mainly ABVD regimen, i.e., Adriamycin, bleomycin, vinblastine and dacarabazine) tailored by PET scan imaging and followed by radiotherapy. In advanced-stage cHL and young patients, ABVD or eBEACOPP (Bleomycin, Etoposide, Doxorubicin, Cyclophosphamide, Vincristine, prednisone and procarbazine) are usually proposed and adapted according to a PET-adapted approach. Recently the combination of Brentuximab vedotin (BV), an antibody–drug conjugate targeting CD30 antigen, with AVD demonstrated a survival benefit compared to standard ABVD, and is likely to become the new standard of care (for a recent review, see [52]) [53]. Specific targeting of the NFkB pathway, either directly or indirectly, is therefore part of this already well-established therapeutic landscape.

Paradoxically, several chemotherapeutic agents, including those used in standard regimens such as doxorubicin, vinblastine, or etoposide, are known to activate NFkB pathways by different mechanisms. However, the effect of the NFkB activation following chemotherapy is cell and context dependent, leading to both pro- and anti-apoptotic effects [54]. Similarly, ionizing radiation (IR) can favor NFkB activation by proteasomal degradation and phosphorylation of IkB by the IKK complex. Importantly, canonical and non-canonical pathways seem to play distinct and antagonistic roles in the mediation of IR sensibility. Indeed, inhibiting the canonical NF-kB pathway dampened the therapeutic effect of ionizing radiation (IR), whereas noncanonical NF-kB deficiency promoted IR-induced anti-tumor immunity in murine models [55]. These data indicate that inhibiting the non-canonical, but not the canonical NFkB pathway could constitute an effective strategy to enhance IR efficacy [55].

Importantly, the link between chemotherapy/radiotherapy and the NFkB pathway could be established by the involvement of the cGAS-STING pathway. Indeed, following chemotherapy or radiotherapy, the release of cytosolic DNA in cancer cells is recognized by Cyclic GMP-AMP (cGAMP) synthase (cGAS), a cytosolic DNA sensor that activates its downstream adaptor, stimulator of interferon genes (STING), located at the endoplasmic reticulum. Then STING signals through three pathways, type I IFN signaling, as well as canonical and non-canonical NF-κB signaling. Notably, the effect of the cGAS/STING pathway in tumor cells is complex and dual, promoting both anti-tumoral immune response and resistance or dissemination [56,57]. Notably, depending on the biological context, its specific involvement in the cHL setting remains to be clarified.

In parallel to these general and non-specific cellular mechanisms, it is possible to consider therapeutic targeting of components of the NFkB.

Proteasomal degradation of IκBα is a key step of the NFkB activation pathway, and subsequently, its inhibition is a potential approach to prevent NFkB activation. The proteasome inhibitor bortezomib has been evaluated as a single agent and in combination with gemcitabine in phase 2 trials, and provided disappointing results in HL [58,59]. However, the in vitro activity of bortezomib against HD-derived cell lines suggests that bortezomib may have therapeutic value for the treatment of HD [60]. However, some data suggest that bortezomib can also be linked to downregulation of IkBα expression, and therefore paradoxically promote NFkB activation in myeloma cells lines and primary tumors cells, suggesting a more complex mechanism of action that may explain the unsuccessful results obtained in HL. Carfilzomib, another related proteasomal blocker, has also been evaluated in several B-cell malignancies, including R/R HL (NCT00150462; NCT02867618) [54,61].

In HRS cell lines and tumors, treatment with an NIK small molecule inhibitor, 4H-isoquinoline-1,3-dione, significantly reduced HL cell viability [9]. This effect appears to be relatively selective, as it is not observed in other B-cell lymphoma cells, which suggests that the noncanonical pathway can be a therapeutic target for HL [9].

Bruton’s tyrosine kinase (BTK) is a member of the TEC family and plays a central role in B-cell signaling, activation, proliferation, and differentiation. BTK is required for the activation of IκB kinase and NFkB in response to BCR engagement. Despite the lack of BCR expression, BTK expression can be observed in approximately 22% of Reed-Sternberg cells. Ibrutinib, a BTK inhibitor, has been successfully used in two heavily pretreated cHL patients, and its efficacy has been confirmed in additional cases [62,63]. A phase II study is underway to evaluate the effectiveness of ibrutinib in RR cHL patients (NCT02824029) [63]. However, the mechanism of ibrutinib activity in cHL remains uncertain, as it seems independent of BTK expression and may implicate Th1-cell polarization of allogenic T cells rather than direct antitumoral activity. Furthermore, exposing HRS cell lines to ibrutinib resulted in the suppression of BTK and other downstream targets, including PI3K, mTOR, and RICTOR, but the levels of total Akt and NF-κB complex component p65 did not significantly decrease [64].

In HRS cell lines and tumors, treatment with an NIK small molecule inhibitor, 4H-isoquinoline-1,3-dione, significantly reduced HL cell viability [9]. This effect appears to be relatively selective, as it is not observed in other B-cell lymphoma cells, which suggests that the noncanonical pathway can be a therapeutic target for HL [9]. Another indirect way of specifically targeting the alternative NFkB pathway is to inhibit Notch signaling. Notch has been identified as an essential upstream regulator of alternative NF-κB signaling, which indicates crosstalk between both pathways in HRS cells. Indeed, Notch signaling inhibition in HRS cells by the γ-secretase inhibitor (GSI) XII results in decreased alternative p52/RelB signaling, interfering with the processing of the NF-κB2 gene product p100 into its active form p52 [65]. Consequently, Notch and NF-κB target genes decreased and impaired HRS cell survival.

Another approach to targeting the NFkB pathway indirectly is to target the PIM protein kinases. Among the common downstream targets of the JAK-STAT and the NF-kB pathways are PIM serine/threonine kinases, which have been shown to be highly expressed in cHL cell lines and primary HRS cells. Targeting of PIM kinases with a pan-PIM inhibitor resulted in tumor cell apoptosis, attenuated JAK-STAT, and NF-kB signaling. Inhibition of PIM kinases downregulated the expression of multiple factors engaged in developing the immunosuppressive microenvironment in cHL, including Gal-1 and PD-L1, and increased the activation of T cells cocultured with RS cells. PIM kinases are therefore promising therapeutic targets in cHL, and inhibitors of such kinases are currently being evaluated in clinical trials [49].

Curcumin, a polyphenol derived from the plant *Curcuma longa*, is known to inhibit NFkB pathway by blocking IKK activity. Curcumin reduces HRS cell growth in vitro, and when given in combination with bleomycin, doxorubicin and vinblastine showed an additive growth inhibitory effect [66]. Interestingly a combination with anti-PD1 monoclonal antibody through nanotechnology display a high therapeutic potential that may be particularly interesting in the HL context [67]. Solid lipid nanoparticles have also been tested for overcoming pharmacokinetic issues associated with this drug [66].

Successful treatment of many cHL patients using antibodies against programmed cell death 1 (PD-1; known as PDCD1) and its ligand (PD-L1; known as CD274) has highlighted the critical importance of PD-1/PD-L1-mediated immune escape in cancer development. If NFkB is emerging as a key positive regulator of PD-L1 expression by both direct (transcriptional) and indirect mechanisms in several solid tumors, its specific and accurate role in HL remains mainly undetermined [68,69]. Indeed, mechanisms leading to the PD-L1 upregulation by the HRS cells are due to several causes, mainly including genetic alterations to the PD-L1 and PDL2 locus of chromosome 9p24.1 (gains, amplifications or fusions), EBV infection that directly activates the PD-L1 promoter via the AP-1/cJUN/JUN-B pathway, and JAK-STAT signaling [4]. However, the extent to which targeting the NFkB pathway could lead to modulation of PDL1 expression by HRS cells and thereby alter the response to ICIs remains to be established.

## 7. Specificities of Nodular Lymphocyte Predominant HL

Many differences distinguish cHL and NLPHL, leading to the consideration of NLPHL as a low-grade B-cell lymphoma. However, as with cHL, GEP demonstrated that the lymphocyte predominant (LP) cells in NLPHL show a strong NFkB activity signature, but this activation is a consequence of different mechanisms than those observed in cHL [70,71].

Contrary to HRS cells, LP cells are EBV negative, CD20 positive, and CD15 negative, and express a functional BCR. Furthermore, LP cells do not express CD30 or BAFF-R, indicating that LP cells may engage other ligand–receptor interactions with the microenvironment that contribute to NF-KB signaling. Genetic lesions are also clearly distinct between the two entities, and do not seem to be a strong driver of NFkB pathway activation. In contrast to HRS cells, TNFAIP3 or NFkBIA are rare genetic events [72], and LP cells share more similarities with T-cell/histiocyte-rich large B-cell lymphoma [73]. Conversely, REL locus amplification is shared by the two entities, whereas others, such as SGK1, DUSP2, and JUNB, appear to be more specific to NLPHL [72,73,74]. Recent work suggests that the NFkB pathway may be activated by a microbial antigen coming from *Moraxella catarrhalis*, a Gram-negative bacterium responsible for respiratory tract infections [75].

## 8. Conclusions

The canonical and alternative NFkB pathways are constitutively activated in cHL by several mechanisms that favor HRS survival and cell interactions to create a favorable microenvironment. This pathway sits at the intersection of complex molecular and cellular mechanisms, including JAK/STAT activation, MAPK activation, EBV infection, immune escape, and crosstalk with the microenvironment. The therapeutic targeting of this pathway appears to be considerably complex and remains a challenge in the era of the well-demonstrated efficacy of monoclonal antibodies.

## Figures and Tables

**Figure 1 biomedicines-10-02153-f001:**
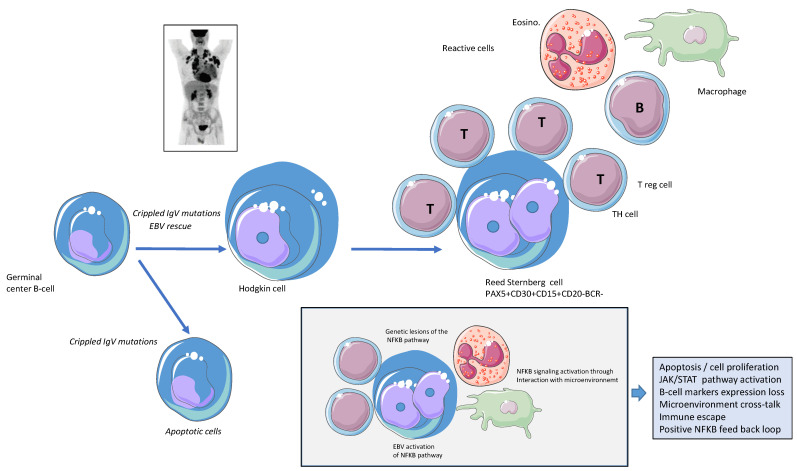
Schematic physiopathological model of classical Hodgkin lymphoma and its relationship with the NFkB pathway. The tumor cells that drive classical Hodgkin lymphoma (cHL) are Hodgkin and Reed-Sternberg (HRS) cells. These B cells are isolated from the germinal center and are rescued from apoptosis despite unproductive or crippled IgV by mechanisms that involve Epstein Barr virus (EBV) or NFkB deregulation. Interaction with the microenvironment, including mainly T cells but also mastocytes, macrophages, and myeloid cells, creates an interaction that favors HRS cell survival. The NFkB pathway is deregulated by multiple mechanisms, including NFkB signaling activation through genetic lesions, interaction with microenvironment cells, and TNF receptor activation or EBV activation. The deregulation of the NFkB pathway is involved in the transcriptional control of several target genes involved in apoptosis, the JAK/STAT pathway, B-cell marker expression loss, microenvironment crosstalk, or cell proliferation. Figure generated using Servier Medical Art and personal archive PET scan imaging.

**Figure 2 biomedicines-10-02153-f002:**
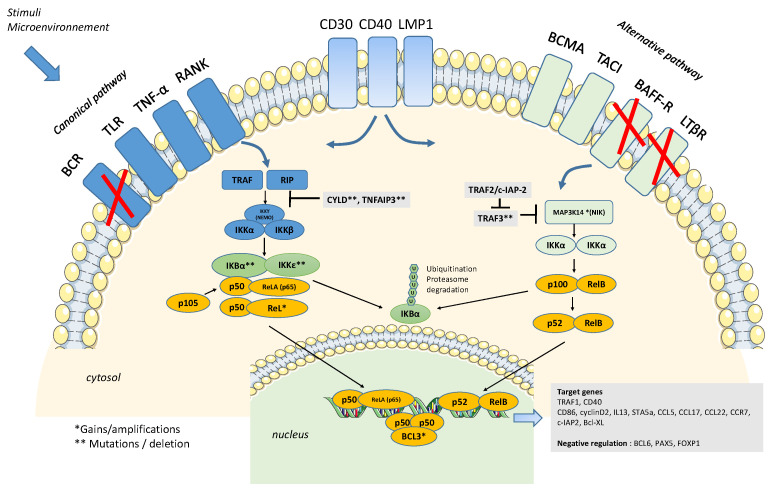
Schematic view of the involvement of the canonical and alternative NFkB pathways in classical Hodgkin lymphoma pathogenesis. The canonical and noncanonical NF-KB pathways are activated in HRS cells by several TNF receptors, except BCR, BAFF-R, and LTB-R, which are not expressed by these cells. CD30, CD40, and LMP1 are able to activate both alternative and canonical pathways. Following receptor activation, TNF receptor-associated factors (TRAFs) interact with the IKK complex or NIK in the canonical and alternative pathways, respectively. This activation leads to the ubiquitination of the inhibitors IKBα and IKBε and their destruction by the proteasome. Transcription factor heterodimers (p50-p65, p52-RelB, p50-p50) are then released and translocated to the nucleus to regulate NFkB target genes (including genes positively or negatively regulated). Negative (A20, CYLD, TRAF3, and IKBα/ε) and positive regulators (Rel, MAP3K14, and BCL3) of the pathways can be targeted by genetic alterations (*/**) in 50% of cHL cases. BCR, BAFF-R and LTβR receptors are usually not expressed by HRS cells. Figure build using Servier Medical Art.

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
