# Peer review of "NFkB Pathway and Hodgkin Lymphoma"

_biomedicines, 2022, doi:10.3390/biomedicines10092153_

Round 1

Reviewer 1 Report

This review is exaustive and well written, and summarizes most of actual knowledge about NFKB pathway deregulation in Hodgkin lymphoma. In spite of the apparent crucial involvement of NFKB factors in HL pathogenesis, drugs inhibiting this pathways have not become so far part of the therapeutic armamentarium of this disease and limited data about their clinical use are available. What do the author think about it?

Author Response

R1. Comments and Suggestions for Authors

This review is exaustive and well written, and summarizes most of actual knowledge about NFKB pathway deregulation in Hodgkin lymphoma. In spite of the apparent crucial involvement of NFKB factors in HL pathogenesis, drugs inhibiting this pathways have not become so far part of the therapeutic armamentarium of this disease and limited data about their clinical use are available. What do the author think about it?

We thank the first referee for these comments. We do agree that usage of specific inhibitors of the NFKB pathways is still limited in cHL. To my opinion the main explanation is the high level of efficacy obtained by standard treatment including immunochemotherapy and radiotherapy that indeed targets directly or indirectly the NFKB pathway. In the revised manuscript, as requested by the editor and other referees we developed more in details this topics in a new formatted chapter (see answer to R3).

Reviewer 2 Report

1.     Why figure 1 is placed behind the figure 2, please correct. 

2.     There are some different fonts in Figure 2 legends, please correct.

3.     Are there Any references for these sentences? 

“cHL is one of the most common lymphomas, with an incidence of 3/100 000 new cases per year in Western countries.” 

“This lymphoma has a bimodal distribution according to age, preferentially affecting adolescents or young adults and those over 60.”

” The tumor cells that drive this neoplasia, namely, Hodgkin and Reed-Sternberg (HRS) cells, display hallmark features that include their rareness in contrast with an extensive and rich reactive microenvironment, their loss of B-cell phenotype markers, their immune escape capacity and the activation of several key biological pathways, including the JAK/STAT signaling, MAPK/ERK and NFK-B pathways”

4.     There are some different fonts in lines 219, 223 and 224. Please correct.

5.     There are some different fonts in conclusion section, lines 263-267. Please correct.

6.     The author should explain the sentence “Recently, the key role of lymphotoxin-a (LTA) as the causative factor for the autocrine and paracrine activation of canonical and noncanonical NFkB has been revealed in cHL cell lines. LTA was isolated from cHL cell lines in the secretome after analysis by chromatography and subsequent mass spectrometry. LTA and its receptor TNFRSF14 are transcriptionally activated by noncanonical NFkB, creating a continuous feedback loop.” more detailed. 

Author Response

R2
We thank the second referee for the comments, corrections and suggestions

  1. Why figure 1 is placed behind the figure 2, please correct. 

This point has been corrected with a switch of the figures.

  1. There are some different fonts in Figure 2 legends, please correct.

This point has been corrected with unification of the fonts are now provided in figure 2

  1. Are there Any references for these sentences? 

These sentences are now referenced as mentioned below:

“cHL is one of the most common lymphomas, with an incidence of 3/100 000 new cases per year in Western countries.” 

“This lymphoma has a bimodal distribution according to age, preferentially affecting adolescents or young adults and those over 60.”

Engert, A. & Younes, A. (Eds.). (2015). Hodgkin,Lymphoma: A Comprehensive Overview, 2nd edn. Springer, Heidelberg.

” The tumor cells that drive this neoplasia, namely, Hodgkin and Reed-Sternberg (HRS) cells, display hallmark features that include their rareness in contrast with an extensive and rich reactive microenvironment, their loss of B-cell phenotype markers, their immune escape capacity and the activation of several key biological pathways, including the JAK/STAT signaling, MAPK/ERK and NFK-B pathways”

Weniger, M.A.; Kuppers, R. Molecular biology of Hodgkin lymphoma. Leukemia 2021, 35, 968-981, doi:10.1038/s41375-021-01204-6

Mottok, A.and Steidl, C. Biology of classical Hodgkin lymphoma: implications for prognosis and novel therapies. Blood. 2018;131(15):1654-1665

  1. There are some different fonts in lines 219, 223 and 224. Please correct.

These points are now corrected

  1. There are some different fonts in conclusion section, lines 263-267. Please correct.

These points are now corrected

  1. The author should explain the sentence “Recently, the key role of lymphotoxin-a (LTA) as the causative factor for the autocrine and paracrine activation of canonical and noncanonical NFkB has been revealed in cHL cell lines. LTA was isolated from cHL cell lines in the secretome after analysis by chromatography and subsequent mass spectrometry. LTA and its receptor TNFRSF14 are transcriptionally activated by noncanonical NFkB, creating a continuous feedback loop.” more detailed. 

This paragraph has been corrected and we provided a more extensive explanation of this point as follow:

Recently, the key role of lymphotoxin-a (LTA) as the causative factor for the autocrine and paracrine activation of canonical and noncanonical NFkB has been revealed in cHL cell lines. LTA was isolated from cHL cell lines in the secretome after analysis by chromatography and subsequent mass spectrometry. Indeed, LTA was the only cytokine isolated from 72 extracellular proteins obtained from the culture supernatant of the 1236 cHL line. The addition of the supernatant or a recombinant LTA molecule leads to the activation of both canonical and noncanonical NFKB pathways in HeLa cells resembling the constitutive activity in cHL. Conversely, the addition of etanerecpt, a neutralizing decoy receptor of LTA, blocks NFKB activation. Furthermore, gene expression data from microdissected cells confirm the high level of expression of LTA by HRS cells, as compared to other B-cell lymphomas. Together, these results lead to the conclusion that LTA is a crucial inducer of the NFKB pathway in cHL.

Reviewer 3 Report

The submitted manuscript focuses on the critical role of NFkB pathway in Hodgkin lymphoma (HL) and the potential targeting of this pathway to improve the outcome of patients with HL. This review has been well written, and the recent literatures were cited appropriately. Nevertheless, there are some sections that need to be discussed in more depth. 

Radiation is one of the pillars for treatment of classic HL either as first or second therapy line.

Therefore, the effect of radiation on NFkB pathway cGAS-STING-dependent or -independent should be discussed. Chemotherapy has critical consequences on NFkB pathway. The contribution of many anti-neoplastic drugs should also be discussed extensively, including and not limited to drugs from ABVD, eBEACOPP, DHAP, ICE/IGEV etc. regimens. In the context of PD-L1 upregulation and the contribution of NFkB for tumor escape, the recent use of anti-PD-1 monoclonal antibodies (pembrolizumab and nivolumab) should be commented. Finally, despite not being used frequently, bortezomib (which partially inhibits NFkB) should also be introduced in the Treatment section.

Minor points: For checkpoint inhibitors abbreviation I would use the accepted one (immune checkpoint inhibitors, ICIs) not Cki which is used in biology to define cyclin-dependent kinase inhibitor. NFkB should be abbreviated as NFkB throughout all manuscript. Software(s) used for drawing the cartoons must be cited (e.g., Servier Medical Art for Figure 2). Moreover, the source of FDG-PET image must be cited.

Author Response

The submitted manuscript focuses on the critical role of NFkB pathway in Hodgkin lymphoma (HL) and the potential targeting of this pathway to improve the outcome of patients with HL. This review has been well written, and the recent literatures were cited appropriately. Nevertheless, there are some sections that need to be discussed in more depth. 

Radiation is one of the pillars for treatment of classic HL either as first or second therapy line.

Therefore, the effect of radiation on NFkB pathway cGAS-STING-dependent or -independent should be discussed. Chemotherapy has critical consequences on NFkB pathway. The contribution of many anti-neoplastic drugs should also be discussed extensively, including and not limited to drugs from ABVD, eBEACOPP, DHAP, ICE/IGEV etc. regimens. In the context of PD-L1 upregulation and the contribution of NFkB for tumor escape, the recent use of anti-PD-1 monoclonal antibodies (pembrolizumab and nivolumab) should be commented. Finally, despite not being used frequently, bortezomib (which partially inhibits NFkB) should also be introduced in the Treatment section.

Minor points: For checkpoint inhibitors abbreviation I would use the accepted one (immune checkpoint inhibitors, ICIs) not Cki which is used in biology to define cyclin-dependent kinase inhibitor. NFkB should be abbreviated as NFkB throughout all manuscript. Software(s) used for drawing the cartoons must be cited (e.g., Servier Medical Art for Figure 2). Moreover, the source of FDG-PET image must be cited.

We thank the third referee for the comments, corrections and suggestions

  • ICI acronym is now used and NFkB is used throughout all manuscript
  • Servier Medical Art is now mentioned, as the PET-FDG imaging source, in the legend of the figure
  • We propose in the revised version a completely new and more detailed  paragraph regarding the interplay between NFKB and cHL treatment, including NFKB targeting y itself but also the impact of the current treatments (chemotherapy, radiation, ICIs) on NFKB.

Targeting the NFKB pathway in cHL

Given the crucial role of the NFKB pathway in the pathophysiology of HL and its involvement in the survival mechanisms of HRS cells, considering the therapeutic and pharmacological targeting of this pathway seems appropriate.

In early stage cHL, the standard of care is the combination of multidrug chemotherapy (mainly ABVD regimen, ie. Adriamycin, bleomycin, vinblastine and dacarabazine) tailored by PET scan imaging and followed by radiotherapy. In advanced-stage cHL and young patients, ABVD or eBEACOPP (Bleomycin, Etoposide, Doxorubicin, Cyclophosphamide, Vincristine, prednisone and  procarbazine) are usually proposed and adapted according to a PET-adapted approach. Recently the combination of Brentuximab vedotin (BV), an antibody-drug conjugate targeting CD30 antigen, with AVD demonstrated a survival benefit as compare to standard ABVD and is likely to become the new standard of care  (for a recent review, see[1])[2].  Specific targeting of the NFKB pathway, either directly or indirectly is therefore part of this already well-established therapeutic landscape.

Paradoxically, several chemotherapeutic agents, including those used in standard regimens such as doxorubicin, vinblastine, or etoposide are known to activate NFKB pathways by different mechanisms. However, the effect of the NFKB activation following chemotherapy is cell and context dependent leading to both pro or anti-apoptotic effects [3]. Similarly, ionizing radiation (IR) can favor NFKB activation by proteasomal degradation and phosphorylation of IkB by the IKK complex. Importantly, canonical and non-canonical pathways seem to play distinct and antagonist roles to mediate IR sensibility. Indeed, inhibiting the canonical NF-kB pathway dampened the therapeutic effect of ionizing radiation (IR), whereas noncanonical NF-kB deficiency promoted IR-induced anti-tumor immunity in murine models [4]. These data indicate that inhibiting the non-canonical, but not the canonical NFKB pathway could constitute an effective strategy to enhance IR efficacy [4].

Importantly, the link between chemotherapy/radiotherapy and the NFKB pathway could be established by the involvement of the cGAS-STING pathway. Indeed, following chemotherapy or radiotherapy the release of cytosolic DNA in cancer cells is recognized by Cyclic GMP-AMP (cGAMP) synthase (cGAS) a cytosolic DNA sensor that activates its downstream adaptor, stimulator of interferon genes (STING), located at the endoplasmic reticulum. Then STING signals through three pathways, type I IFN signaling, as well as canonical and non-canonical NF-κB signaling. Of note the effect of cGAS/STING  pathway in tumor cells is complex and dual by promoting both anti-tumoral immune response and resistance or  dissemination  [5,6]. Of note, depending of the biological context, its specific involvement in the cHL setting remains to be clarify.

In parallel to these general and non-specific cellular mechanisms, it is possible to consider a therapeutic targeting of components of the NFKB.

Proteasomal degradation of IkBα is a key-step of the NFKB activation pathway and subsequently its inhibition a potential approach to prevent NFKB activation. Proteasome inhibitors bortezomib has been evaluated as single agent or in combination with gemcitabine in phase 2 trials and provided disappointing results HL.[7,8]. However, the in vitro activity of bortezomib against HD-derived cell lines suggests that bortezomib may have a therapeutic value for the treatment of HD[9]. However, some data suggest that bortezomib can be also linked to downregulation of IkBα expression and therefore paradoxically promoting NFKB activation in myeloma cells lines and primary tumors cells, suggesting a more complex mechanism of action that may explain the unsuccessful results obtained in HL.Carfilzomib, an other related proteasomal blocker, is also evaluated in several B-cell malignancies including R/R HL (NCT00150462; NCT02867618) [3,10].

In HRS cell lines and tumors, treatment with a NIK small molecule inhibitor, 4H-isoquinoline-1,3-dione, significantly reduced HL cell viability.[11] This effect appears to be relatively selective as it is not observed in other B-cell lymphoma cells, which suggests that the noncanonical pathway can be a therapeutic target for HL[11].

Bruton's tyrosine kinase (BTK) is a member of the TEC family and plays a central role in B-cell signaling, activation, proliferation, and differentiation. BTK is required for the activation of IκB kinase and NFKB in response to BCR engagement. Despite the lack of BCR expression, BTK expression can be observed in approximately 22% of Reed-Sternberg cells. Ibrutinib, a BTK inhibitor, has been successfully used in two heavily pretreated cHL patients, and its efficacy has been confirmed in additional cases[12,13] . A phase II study is underway to evaluate the effectiveness of ibrutinib in RR cHL patients (NCT02824029)[13]. However, the mechanism of ibrutinib activity in cHL remains uncertain, as it seems independent of BTK expression and may implicate Th1-cell polarization of allogenic T cells rather than direct antitumoral activity. Furthermore, exposing HRS cell lines to ibrutinib resulted in the suppression of BTK and other downstream targets, including PI3K, mTOR, and RICTOR, but the levels of total Akt and NF-κB complex component p65 did not significantly decrease[14].

In HRS cell lines and tumors, treatment with a NIK small molecule inhibitor, 4H-isoquinoline-1,3-dione, significantly reduced HL cell viability.[11] This effect appears to be relatively selective as it is not observed in other B-cell lymphoma cells, which suggests that the noncanonical pathway can be a therapeutic target for HL[11]. Another indirect way to specifically target the alternative NFKB pathway is to inhibit Notch signaling. Notch has been identified as an essential upstream regulator of alternative NF-κB signaling, which indicates crosstalk between both pathways in HRS cells. Indeed, Notch signaling inhibition in HRS cells by the γ-secretase inhibitor (GSI) XII results in decreased alternative p52/RelB signaling, interfering with the processing of the NF-κB2 gene product p100 into its active form p52[15]. Consequently, Notch and NF-κB target genes decreased and impaired HRS cell survival.

Another approach to targeting the NFKB pathway indirectly is to target the PIM protein kinases. Among the common downstream targets of the JAK-STAT and the NF-kB pathways are PIM serine/threonine kinases, which have been shown to be highly expressed in cHL cell lines and primary HRS cells. Targeting of PIM kinases with a pan-PIM inhibitor resulted in tumor cell apoptosis, attenuated JAK-STAT and NF-kB signaling. Inhibition of PIM kinases downregulated expression of multiple factors engaged in developing the immunosuppressive microenvironment in cHL, including Gal-1 and PD-L1, and increased activation of T cells cocultured with RS cells. PIM kinases are therefore promising therapeutic targets in cHL and such kinases inhibitors are currently evaluated in clinical trials [16].

Curcumin, a polyphenol derived from the plant Curcuma longa, is known to inhibit NFKB pathway by blocking IKK activity. Curcumin reduces HRS cell growth in vitro, and when given in combination with bleomycin, doxorubicin and vinblastine showed an additive growth inhibitory effect [17]. Interestingly a combination with anti-PD1 monoclonal antibody through nanotechnology display a high therapeutic potential that may be particularly interesting in HL context [18]. Solid lipid nanoparticles has been also tested to overcome pharmacokinetics issues associated to this drug [17].

 Successful treatment of many cHL patients using antibodies against programmed cell death 1 (PD-1; known as PDCD1) and its ligand (PD-L1; known as CD274) has highlighted the critical importance of PD-1/PD-L1-mediated immune escape in cancer development.  If NFKB is emerging as a key positive regulator of PD-L1 expression by both direct (transcriptional) and indirect mechanisms in several solid tumors, its specific and accurate role in HL remains mainly undetermined [19,20]. Indeed, mechanisms leading to the PD-L1 upregulation by the HRS cells are due to several causes including mainly genetic alterations to the PD-L1 and PDL2 locus of chromosome 9p24.1 (gains, amplifications or fusions), EBV infection that  directly activates the PD-L1 promoter via the AP-1/cJUN/JUN-B pathway and JAK-STAT signaling[21]. However, the extent to which targeting the NFKB pathway could lead to modulation of PDL1 expression by HRS cells and thereby alter the response to ICIs remains to be established.

 .

References

  1. Mohty, R.; Dulery, R.; Bazarbachi, A.H.; Savani, M.; Hamed, R.A.; Bazarbachi, A.; Mohty, M. Latest advances in the management of classical Hodgkin lymphoma: the era of novel therapies. Blood Cancer J 2021, 11, 126, doi:10.1038/s41408-021-00518-z.
  2. Ansell, S.M.; Radford, J.; Connors, J.M.; DÅ‚ugosz-Danecka, M.; Kim, W.S.; Gallamini, A.; Ramchandren, R.; Friedberg, J.W.; Advani, R.; Hutchings, M.; et al. Overall Survival with Brentuximab Vedotin in Stage III or IV Hodgkin's Lymphoma. N Engl J Med 2022, 387, 310-320, doi:10.1056/NEJMoa2206125.
  3. Puar, Y.R.; Shanmugam, M.K.; Fan, L.; Arfuso, F.; Sethi, G.; Tergaonkar, V. Evidence for the Involvement of the Master Transcription Factor NF-kappaB in Cancer Initiation and Progression. Biomedicines 2018, 6, doi:10.3390/biomedicines6030082.
  4. Hou, Y.; Liang, H.; Rao, E.; Zheng, W.; Huang, X.; Deng, L.; Zhang, Y.; Yu, X.; Xu, M.; Mauceri, H.; et al. Non-canonical NF-κB Antagonizes STING Sensor-Mediated DNA Sensing in Radiotherapy. Immunity 2018, 49, 490-503.e494, doi:10.1016/j.immuni.2018.07.008.
  5. Gao, M.; He, Y.; Tang, H.; Chen, X.; Liu, S.; Tao, Y. cGAS/STING: novel perspectives of the classic pathway. Mol Biomed 2020, 1, 7, doi:10.1186/s43556-020-00006-z.
  6. Ng, K.W.; Marshall, E.A.; Bell, J.C.; Lam, W.L. cGAS-STING and Cancer: Dichotomous Roles in Tumor Immunity and Development. Trends Immunol 2018, 39, 44-54, doi:10.1016/j.it.2017.07.013.
  7. Blum, K.A.; Johnson, J.L.; Niedzwiecki, D.; Canellos, G.P.; Cheson, B.D.; Bartlett, N.L. Single agent bortezomib in the treatment of relapsed and refractory Hodgkin lymphoma: cancer and leukemia Group B protocol 50206. Leuk Lymphoma 2007, 48, 1313-1319, doi:10.1080/10428190701411458.
  8. Mendler, J.H.; Kelly, J.; Voci, S.; Marquis, D.; Rich, L.; Rossi, R.M.; Bernstein, S.H.; Jordan, C.T.; Liesveld, J.; Fisher, R.I.; et al. Bortezomib and gemcitabine in relapsed or refractory Hodgkin's lymphoma. Ann Oncol 2008, 19, 1759-1764, doi:10.1093/annonc/mdn365.
  9. Zheng, B.; Georgakis, G.V.; Li, Y.; Bharti, A.; McConkey, D.; Aggarwal, B.B.; Younes, A. Induction of cell cycle arrest and apoptosis by the proteasome inhibitor PS-341 in Hodgkin disease cell lines is independent of inhibitor of nuclear factor-kappaB mutations or activation of the CD30, CD40, and RANK receptors. Clin Cancer Res 2004, 10, 3207-3215, doi:10.1158/1078-0432.ccr-03-0494.
  10. Hideshima, T.; Ikeda, H.; Chauhan, D.; Okawa, Y.; Raje, N.; Podar, K.; Mitsiades, C.; Munshi, N.C.; Richardson, P.G.; Carrasco, R.D.; et al. Bortezomib induces canonical nuclear factor-kappaB activation in multiple myeloma cells. Blood 2009, 114, 1046-1052, doi:10.1182/blood-2009-01-199604.
  11. Ranuncolo, S.M.; Pittaluga, S.; Evbuomwan, M.O.; Jaffe, E.S.; Lewis, B.A. Hodgkin lymphoma requires stabilized NIK and constitutive RelB expression for survival. Blood 2012, 120, 3756-3763, doi:10.1182/blood-2012-01-405951.
  12. Hamadani, M.; Balasubramanian, S.; Hari, P.N. Ibrutinib in Refractory Classic Hodgkin's Lymphoma. N Engl J Med 2015, 373, 1381-1382, doi:10.1056/NEJMc1505857.
  13. Badar, T.; Astle, J.; Kakar, I.K.; Zellner, K.; Hari, P.N.; Hamadani, M. Clinical activity of ibrutinib in classical Hodgkin lymphoma relapsing after allogeneic stem cell transplantation is independent of tumor BTK expression. Br J Haematol 2020, 190, e98-e101, doi:10.1111/bjh.16738.
  14. Muqbil, I.; Chaker, M.; Aboukameel, A.; Mohammad, R.M.; Azmi, A.S.; Ramchandren, R. Pre-clinical anti-tumor activity of Bruton's Tyrosine Kinase inhibitor in Hodgkin's Lymphoma cellular and subcutaneous tumor model. Heliyon 2019, 5, e02290, doi:10.1016/j.heliyon.2019.e02290.
  15. Schwarzer, R.; Dorken, B.; Jundt, F. Notch is an essential upstream regulator of NF-kappaB and is relevant for survival of Hodgkin and Reed-Sternberg cells. Leukemia 2012, 26, 806-813, doi:10.1038/leu.2011.265.
  16. Szydlowski, M.; Prochorec-Sobieszek, M.; Szumera-Cieckiewicz, A.; Derezinska, E.; Hoser, G.; Wasilewska, D.; Szymanska-Giemza, O.; Jablonska, E.; Bialopiotrowicz, E.; Sewastianik, T.; et al. Expression of PIM kinases in Reed-Sternberg cells fosters immune privilege and tumor cell survival in Hodgkin lymphoma. Blood 2017, 130, 1418-1429, doi:10.1182/blood-2017-01-760702.
  17. Guorgui, J.; Wang, R.; Mattheolabakis, G.; Mackenzie, G.G. Curcumin formulated in solid lipid nanoparticles has enhanced efficacy in Hodgkin's lymphoma in mice. Arch Biochem Biophys 2018, 648, 12-19, doi:10.1016/j.abb.2018.04.012.
  18. Xiao, Z.; Su, Z.; Han, S.; Huang, J.; Lin, L.; Shuai, X. Dual pH-sensitive nanodrug blocks PD-1 immune checkpoint and uses T cells to deliver NF-kappaB inhibitor for antitumor immunotherapy. Sci Adv 2020, 6, eaay7785, doi:10.1126/sciadv.aay7785.
  19. Antonangeli, F.; Natalini, A.; Garassino, M.C.; Sica, A.; Santoni, A.; Di Rosa, F. Regulation of PD-L1 Expression by NF-κB in Cancer. Front Immunol 2020, 11, 584626, doi:10.3389/fimmu.2020.584626.
  20. Betzler, A.C.; Theodoraki, M.N.; Schuler, P.J.; Döscher, J.; Laban, S.; Hoffmann, T.K.; Brunner, C. NF-κB and Its Role in Checkpoint Control. Int J Mol Sci 2020, 21, doi:10.3390/ijms21113949.
  21. Mottok, A.; Steidl, C. Biology of classical Hodgkin lymphoma: implications for prognosis and novel therapies. Blood 2018, 131, 1654-1665, doi:10.1182/blood-2017-09-772632.

Round 2

Reviewer 3 Report

Dear author and editor,

I would like to congratulate the author for taking very serious all reviewer's comments. from my perspective, the manuscript is publishable in the present form.

Thank you.